# Three-Dimensional-Printed Modular Titanium Alloy Plates for Osteosynthesis of the Jawbone

**DOI:** 10.3390/biomedicines12071466

**Published:** 2024-07-02

**Authors:** Maciej Dobrzyński, Maria Szymonowicz, Joanna Nowicka, Magdalena Pajączkowska, Anna Nikodem, Piotr Kuropka, Magdalena Wawrzyńska, Agnieszka Rusak

**Affiliations:** 1Department of Pediatric Dentistry and Preclinical Dentistry, Wroclaw Medical University, 50-425 Wroclaw, Poland; maciej.dobrzynski@umw.edu.pl; 2Pre-Clinical Research Centre, Wroclaw Medical University, 50-368 Wroclaw, Poland; maria.szymonowicz@umw.edu.pl (M.S.); magdalena.wawrzynska@umw.edu.pl (M.W.); 3Department and Division of Microbiology, Faculty of Medicine, Wroclaw Medical University, 50-556 Wroclaw, Poland; joanna.nowicka@umw.edu.pl (J.N.); magdalena.pajaczkowska@umw.edu.pl (M.P.); 4Department of Mechanics, Materials and Biomedical Engineering, Faculty of Mechanical Engineering, Wroclaw University of Science and Technology, 50-371 Wroclaw, Poland; anna.nikodem@pwr.edu.pl; 5Division of Histology and Embryology, Department of Biostructure and Animal Physiology, Faculty of Veterinary Medicine, Wroclaw University of Environmental and Life Sciences, 50-366 Wroclaw, Poland; piotr.kuropka@upwr.edu.pl; 6Division of Histology and Embryology, Department of Human Morphology and Embryology, Faculty of Medicine, Wroclaw Medical University, 50-368 Wroclaw, Poland

**Keywords:** titanium alloy, 3D printing, CAD, modular plate, biocompatibility, biomaterial

## Abstract

Backgrounds: The titanium–aluminum–vanadium alloy (Ti-6Al-4V) is frequently used in implantology due to its biocompatibility. The use of 3D printing enables the mechanical modification of implant structures and the adaptation of their shape to the specific needs of individual patients. Methods: The titanium alloy plates were designed using the 3D CAD method and printed using a 3D SLM printer. Qualitative tests were performed on the material surface using a microcomputed tomography scanner. The cytotoxicity of the modular titanium plates was investigated using the MTT assay on the L929 cell line and in direct contact with Balb/3T3 cells. Cell adhesion to the material surface was evaluated with hFOB1.19 human osteoblasts. Microbial biofilm formation was investigated on strains of *Lactobacillus rhamnosus*, *Staphylococcus epidermidis*, *Streptococcus mutans* and *Candida albicans* using the TTC test and scanning electron microscopy (SEM). Results: The surface analysis showed the hydrophobic nature of the implant. The study showed that the titanium plates had no cytotoxic properties. In addition, the material surface showed favorable properties for osteoblast adhesion. Among the microorganisms tested, the strains of *S. mutans* and *S. epidermidis* showed the highest adhesion capacity to the plate surface, while the fungus C. albicans showed the lowest adhesion capacity. Conclusions: The manufactured modular plates have properties that are advantageous for the implantation and reduction in selected forms of microbial biofilm. Three-dimensional-printed modular titanium plates were investigated in this study and revealed the potential clinical application of this type of materials, regarding lack of cytotoxicity, high adhesion properties for osteoblasts and reduction in biofilm formation. The 3D CAD method allows us to personalise the shape of implants for individual patients.

## 1. Introduction

Titanium alloys are a commonly used material for the production of 3D-printed medical implants [1]. Using 3D printing, implants can be customized and personalized to meet the unique needs of each patient [2,3]. The design of these implants can be printed and customized based on anatomical data from computed tomography or magnetic resonance imaging [2]. This results in an exact reproduction of the natural shape of hard tissues [4]. Titanium products are used to stabilize bone fragments, including plates for osteosynthesis of facial bones [5]. Modular plates for osteosynthesis of the mandibular bone can also be manufactured, which are characterized by individual fit and universality and enable better axial stabilization of bone fragments. The prototype of the manufactured modular plates fulfills the same osteosynthesis parameters as the standard plates. The SLM production technology can be successfully used to manufacture osteosynthesis plates from biocompatible metal alloys.

In 3D printing techniques for metal powder, such as selective laser melting (SLM) and selective electron beam melting (SEBM), metal powder is melted with a high-energy beam and deposited layer by layer to create the desired shape of the metal implant [1,2,3,6] The titanium–aluminum–vanadium alloy (Ti-6Al-4V) has the advantage that it has a low thermal conductivity coefficient (21.4 W/mK), which helps to protect the bone tissue from thermal irritation. It is characterized by a low density (4.5 g/cm^3^), which results in a relatively low mass of the manufactured materials, while maintaining adequate stiffness and high mechanical strength (up to 1800 MPa) [7,8]. This is particularly important for dental materials that are exposed to high loads. Titanium implants have several advantageous properties, including high transmittance of X-rays, which prevents interference when reading radiologic images such as panoramic radiographs, radiovisiography and CBCT (cone beam computed tomography). In addition, titanium is a non-magnetic material, which is desirable as ferromagnetic phases can cause thrombotic effects [6,7]. Titanium materials consist either of pure titanium or of alloys containing vanadium and aluminum and occasionally other elements. Pure titanium is highly biocompatible, but has lower strength and flexibility. Alloys with vanadium and aluminum, on the other hand, are more resistant to physical loads. Therefore, titanium alloys are more commonly used for the production of biomaterials [4,8,9]. The Ti-6Al-4V alloy used in this study contains about 6% aluminum and 4% vanadium.

In maxillofacial traumatology and orthognathic surgery, prefabricated osteotomy plate patterns are often used to rigidly connect bone fragments. Each fracture or reconstructive procedure is a unique clinical case, and there is not always an optimal geometry or shape of the osteotomy plate for a given situation. In addition, the difficult intraoral anatomical access within the facial skeleton often requires modification of the original shape or size by bending or cutting. However, such modifications may compromise the stability of the connection and the biocompatibility of the plate. The fabrication of a universal plate for the osteosynthesis of mandibular bones is a crucial aspect of trauma surgery. In some cases, the geometries or shapes of osteotomy plates are not optimal, so the original shape or size must be modified by bending or cutting. Such modifications can affect the stability and durability of the connection.

The present work presents an innovative utility model for a modular osteotomy plate consisting of two or more submodules. This allows physicians to connect multiple plate modules together in real time to design and create the required shapes in the operating room. An important problem with implants is the formation of a biofilm on their surface, which can lead to infections [10,11]. It is therefore important to look for biomaterials that integrate well into the surrounding tissue and are resistant to microbial adhesion.

The aim of the study was to determine the cytotoxicity of modular osteosynthesis plates made of a titanium alloy using 3D printing technology and the susceptibility of the biomaterial to microbial adhesion and biofilm formation. Evaluation of the biocompatibility and cytotoxicity of a plate made of TiV powder sintered with a high-power laser used 3D SLM printers in the production of plates made of biocompatible metal alloys.

## 2. Materials and Methods

### 2.1. Titanium Modular Plates

Samples of the titanium alloy Ti6Al4V in the form of stars were used for the tests (Figure 1). The plates were manufactured in cooperation with ScienceBioTech Ltd. (Wrocław, Poland), which has been granted utility model protection (Nos. 003434067-0001, 004567717-0001) and a patent (P.424141). The prototype of the circuit board was created in a 3D CAD program and then printed using an SLM (selective laser melting) 3D printer (Materialise, Leuven, Belgium). The samples for the biological tests were sterilized in an autoclave at a temperature of 134 °C in a SterilClave 18B sterilizer (Milan, Italy).

### 2.2. Analysis of Titanium Modular Plate Composition

EDS analysis was carried out based on the SEM measurement. Scanning electron microscopy (SEM) coupled with energy-dispersive X-ray diffraction (EDS) (Phenom ProX; Thermo Scientific, Waltham, MA, USA) was used. The elemental composition as evaluated by EDS was analyzed based on weight and atomic percentage for all samples. The EDS spectra of an example sample together with its SEM image and the atomic composition of Ti6Al4V titanium alloy are shown in Figure 2.

### 2.3. Analysis of the Structure and Topography

Qualitative studies of the material surface were carried out using an 1172 SkyScan microcomputed tomography system, Bruker (Kontich, Belgium), with an image resolution of 6 μm. The lamp parameters were set to 100 kV/100 μA, with a unit rotation angle of 0.4° and total angle 360°. All reconstructions were obtained using NRecon software (Version 1.7.1.0) and the images shown were presented using CTVox (Bruker, Kontich, Belgium). Differential analysis of the plate surface after bacterial and fungal deposition was also performed using micro-computed tomography. Differential analysis is a tool in which, by comparing the reconstruction of plates before and after incubation of fungi and bacteria, it is possible to determine surface changes in the material.

To obtain the surface topography of the implants, a Leica DCM8 optical profilometer (Leica Microsystems, Wetzlar, Germany) was used in the measurements. Measurements were taken at several locations on the 600 × 800 μm implant attachment surface using a confocal microscope, with 50× magnification. The surface images obtained by the profilometer were analysed in Leica MAP Software to derive values for topography, roughness and waviness parameters.

### 2.4. Analysis of the Contact Angle

The contact angle measurement was carried out with the Surftens Universal device from OEG (Frankfurt, Germany). The static contact angle was measured for a 0.8 μL water droplet at 22 °C.

### 2.5. Evaluation of the Roughness

The roughness was analyzed using a HOMMEL-ETAMIC W5 roughness meter from Jenoptik (Villingen-Schwenningen, Germany). The measuring distance (lt = 1.5 mm) and the elementary distance (lc = 0.25 mm) were measured at a feed rate (vt = 0.15 mm/s) in accordance with the DIN EN ISO 11562 standard [12].

### 2.6. Cell Culture

The normal L929 and Balb/3T3 mouse cell lines (ATCC, American Type Culture Collection ATCC, Old Town Manassas, VA, USA) were grown in EMEM medium (Lonza, Basel, Switzerland) and in DMEM medium containing 4.5 g/mL glucose. The media were supplemented with 1% L-glutamine with streptomycin and penicillin (Sigma-Aldrich, St. Louis, MO, USA) and 10% FBS (fetal bovine serum, Sigma-Aldrich). Human osteoblast cells of the hFOB1.19 line (ATCC) were cultured in a 1:1 mixture of DMEM (Lonza) and F12K (ATCC) medium. Cultivation was performed under the following standard conditions: 37 °C, 5% CO_2_, at constant humidity (HERAcell 150i, Thermo Scientific, Waltham, MA, USA). Cells were passaged at 60–70% confluence and the medium was changed 3 times per week. Cytotoxicity tests were performed with the normal mouse fibroblast lines L929 (ATCC) and Balb/3T3 (ATCC), which are reference lines in the in vitro evaluation of cytotoxicity of biomaterials according to ISO 10993-5:2009 [13,14,15,16], as well as with normal human osteoblasts hFOB1.19.

#### 2.6.1. Direct Contact

Balb/3T3 cells were seeded in a 6-well plate (TPP) with 1.5 × 10^5^ cells per well in 3 mL of medium. After 24 h of incubation at 37 °C, the tested material was plated on a monolayer of cells in each well. Cell culture was then shuttled for 24 h, which allowed evaluation of the adhesion properties of the disks for Balb/3T3 cells. The control in the study was cells without contact with the material. After 24 h, the morphology of the cells was evaluated. Cell morphology was examined using a CKX53 inverted phase contrast microscope (Olympus, Tokyo, Japan) [15,17,18].

#### 2.6.2. Evaluation of Cell Morphology

The degree of toxicity upon direct contact of the cells with the tested material was assessed on a scale of 0–4, according to the ISO 10993:5 standard [16] and based on changes in the morphology of the fibroblast cells. The cell morphology was evaluated taking into account the occurrence of lysis, changes in cell shape and the degree of confluence of the culture compared to the control culture. No toxicity (grade 0) is described as no changes under or near the tested material. The grading of toxicity is as follows: 1—weak degree of reactivity; individual cells have degenerated or deformed under the material; 2—moderate degree of reactivity; the zone of altered cells is limited to the surface under the material; 3—moderate degree of reactivity; the zone of altered cells is limited to 1 cm around the material; 4—strong degree of reactivity; the zone of altered cells exceeds the 1 cm limit around the material. Changes in cultures above the 2nd degree and a reduction in the cell survival rate of more than 30% are criteria that prove the cytotoxic properties of the tested materials [13,18].

#### 2.6.3. Preparation of Extracts

The extracts from the material were prepared at a ratio of 0.2 g/mL according to ISO 10993:12 standard [19] guidelines for sample preparation and reference materials. Extraction was performed in complete DMEM medium (Lonza) at 37 °C for 24 h. Extracts at concentrations of 100%, 50%, 25% and 12.5% were used for the tests. Sodium lauryl sulfate (SLS, Sigma-Aldrich) solutions at concentrations of 0.2, 0.1 and 0.05 mg/mL served as a positive control, while high-density polyethylene (HDPE, U.S. Pharmacopeia—Rockville, MD, USA) was used as a negative control. The blank was the complete culture medium [13,14].

#### 2.6.4. MTT Assay

L929 fibroblast cells were seeded in a 96-well plate (TPP, Trasadingen, Switzerland) with 1 × 10⁴ cells per well. After 24 h, the medium was removed and 100 µL of the extracts of the tested material (100%, 50%, 25% and 12.5%), SLS solutions (positive control) and HDPE extract as the negative control were added to the wells. After 24 h of incubation, the MTT test was performed. Next, 50 µL of a 1 mg/mL MTT solution was added to each well [11]. After 2 h of incubation, 100 μL of DMSO (Sigma-Aldrich) was added. The absorbance was measured at a wavelength of 570 nm using the ELx800 spectrophotometer and Gen5 software (BioTek, Winooski, VT, USA) [13,14]. The cell survival rate (V%) was calculated according to Equation (1).
V% = (Ab − Am/Ac − Am) × 100,(1)

In the above equation, Ab is the absorbance of the test sample, Am is the absorbance of the medium and Ac is the absorbance of the control sample.

#### 2.6.5. Cell Adhesion

Human hFOB1.19 osteoblasts were trypsinized (TrypLE, Gibco, Waltham, MA, USA), suspended in DMEM/F12K complete culture medium (1:1) and seeded in 6-well plates (TPP) at 8.0 × 10^4^ cells/well. Before the culture was established, the tested material was added to each plate. The cells were then cultured for 48 h in a gentle rocking motion, which allowed the evaluation of the adhesion properties of the materials to the cells. The control in the study was cells without contact with the material. After the incubation period, cell adhesion was assessed with the fluorescent dyes DAPI (0.1 µg/mL, Thermo Fisher, Waltham, MA, USA) and propidium iodide (0.5 mg/mL, Roche, Germany), (DAPI/Rhodamine), using an Eclipse80i microscope (Nikon Corporation, Japan) [15,17].

### 2.7. Quantitative Evaluation of Bacterial Adhesion

The following reference strains were selected for this work: *Streptococcus mutans* (ATCC 25175), *Lactobacillus rhamnosus* (ATCC 9595), *Candida albicans* (ATCC 90028) and the reference strain *Staphylococcus epidermidis* RP62A (ATCC 35984) with strong biofilm-forming capabilities.

A suspension of microorganisms was prepared from fresh cultures of the investigated strains, corresponding to a density of 0.5 on the McFarland scale (1.5 × 10^6^ CFU/mL) and 1.0 (3 × 10^8^ CFU/mL) in the case of bacteria and fungi, respectively. To obtain a suspension of the strains, Sabouraud Dextrose Broth liquid medium (Biomaxima, Lublin, Poland), Brain Heart Infusion Broth liquid medium (BHI) (Biomaxima, Lublin, Poland), Man–Rogosa–Sharpe Broth medium (MRS) (Biomaxima, Lublin, Poland) and Tryptic Soy Broth (TSB) (Biomaxima, Lublin, Poland) tested for *Candida albicans*, *Streptococcus mutans*, *Lactobacillus rhamnosus* and *Staphylococcus epidermidis* were used.

Sterile materials were added to the prepared microorganism suspension. After the incubation period (*S. mutans*—37 °C 48 h in CO_2_ atmosphere; *L. rhamnosus*—37 °C 48 h under anaerobic conditions; *C. albicans*—37 °C 48 h under aerobic conditions; *S. epidermidis*—37 °C 24 h under aerobic conditions), the materials were washed three times in NaCl and shaken for 1 min in 1 mL of 0.5% saponin solution (Sigma-Aldrich). The obtained suspension of microorganisms desorbed from the surface of the material was quantitatively inoculated onto solid media suitable for a given microorganism. After the incubation period, the grown colonies were counted and the number of colony-forming units per 1 mL of the suspension (CFU/mL) was determined. The CFU/mL value was calculated according to Formula (2). The study was performed in triplicate [14,18].
CFU/mL = average colony count × reciprocal of dilution × 10(2)

### 2.8. Qualitative Assessment of Bacterial Biofilm Formation

For the qualitative evaluation of bacterial biofilm formation, a bacterial suspension with a density of 0.5 on the McFarland scale (1.5 × 10^8^ CFU/mL) was prepared from an 18 h culture of reference strains. The tooth material was then added to the suspension and the mixture was incubated. After the incubation period, the biomaterial was washed and transferred to a liquid medium with the addition of 1% (10 mg/mL) 2,3,5-triphenyltetrazolium chloride (TTC). After the incubation period, the samples were washed and the presence of red formazan produced as a result of TTC reduction by live microorganisms was evaluated. The degree of TTC reduction was determined using the following 5-point scale: 0—no TTC reduction, 1 (+)—localized reddening of the biomaterial, 2 (++)—the surface is slightly reddened, 3 (+++)—the entire surface is reddened and the substrate is pink; 4 (++++)—the entire surface and bottom are strongly reddened.

An 18 h culture of *C. albicans* was used to prepare a suspension with a density of 0.5 on the McFarland scale (1.5 × 10^6^ CFU/mL). The dental material was then added to the suspension and incubated under aerobic conditions at 37 °C for 48 h. After the incubation period, the biomaterial was washed and transferred to a liquid medium with the addition of 3-[4,5-dimethylthiazol-2-yl]-2,5-diphenyltetrazolium bromide (MTT) and incubated under aerobic conditions at 37 °C for a further 24 h. The biomaterial was then analyzed for MTT reduction using a scoring system from 0 to 4, where 0 indicates no reduction, while 4 indicates a strong reduction in which the entire surface of the material and substrate appears purple. Values from 1 to 3 indicate varying degrees of reduction, with purple patches or partial coverage of the surface [13,14,18].

### 2.9. Scanning Electron Microscopy (SEM)

A suspension with a density of 0.5 McFarland in liquid Sabouraud, BHI, MRS and TSB medium was prepared from fresh cultures of the investigated strains for *C. albicans*, *S. mutans*, *L. rhamnosus* and *S. epidermidis*, respectively. The samples were incubated for 24 h at 37 °C under aerobic conditions, in a CO_2_ atmosphere and under anaerobic conditions with shaking. After the incubation period, 100 μL of the suspension was transferred to the corresponding well of a 12-well plate, fixed, sprayed with gold and examined with a ZEISS scanning electron microscope model EVO LS15 (Carl Zeiss, Carl Zeiss, Jena, Germany).

### 2.10. Statistical Analysis

The Shapiro–Wilk test was used to analyze the normal distribution. One-way ANOVA with Tukey’s post hoc tests for multiple comparisons was used for statistical analysis. Statistical significance was defined as a level of *p* < 0.05. Statistical analysis was performed using Statistica 13.3 software (Tibco, Palo Alto, CA, USA).

## 3. Results

### 3.1. Modular Titanium Plates Showed Hydrophobic Surface Properties

One of the extremely important parameters determining the biological response of implants (with particular focus on their osteointegration) is primarily the surface development. This development, from the point of view of cell adhesion and tissue (protein-surface) junction formation, depends on both the hydration of the surface and its profile related to surface roughness and waviness. The analyzed surface structure of the tested material is shown in Figure 3.

The surface area of the implant at the flattest point (attachment part), where the contact angle and roughness were measured, is approximately 10.5 mm^2^. The implant has a width of 2.8 mm and a height of 3.3 mm. The average static contact angle for the tested samples is 96.64° ± 5.95° and is in the range of 84.9–106°. This angle value indicates the hydrophobic nature of the implant. In the study, tests were performed on a flat fragment, specifically on the lower part of the implant.

Roughness refers to the fine-scale irregularities that occur over short wavelengths, while waviness refers to the larger-scale variations that occur over longer wavelengths (Figure 4). The values of the texture and tomography parameters (topography, roughness and surface waviness), measured according to ISO 25178 [20], are summarised in Table 1. The surface texture parameters were evaluated by measuring the following values: Sa, Sq, Sz, Sz, Ssk, Sku, Sp; Sv. The results obtained are shown in Table 1. The table shows the separated results of the texture parameters for topography, roughness and surface waviness, measured according to ISO 25178.

### 3.2. Modular Titanium Plates Showed No Cytotoxic Effects on Fibroblast Cells

Upon direct contact, the morphology of Balb/3T3 cells was observed under the modular plates, near the material and in the rest of the culture. No typical changes associated with cytotoxic effects, such as cell shrinkage, lysis or inhibition of culture growth, were observed (see Figure 5). The results of the MTT test show that the modular titanium plates do not exhibit cytotoxicity towards L929 cells. The cell survival rate did not fall below 100% for any of the extracts tested.

In addition, a stimulating effect on fibroblasts was observed in both the 100% and 50% extracts. Results of the MTT assay after 24 h of contact of L929 cells with the tested extracts showed statistically significant higher cell viability for 100% compared to the control (*p* = 0.000013) and 50% compared to the control (*p* = 0.0024). Cell morphology appeared normal in all samples tested. The positive, negative and blank controls showed normal reactivity of the L929 cell culture (Figure 5). The positive controls, SLS extracts, showed statistically significant decreased cell viability compared to control (*p* = 0.000012) (Figure 5).

### 3.3. Cell Adhesion on the Surface of the Plates

After 48 h of incubation, adhesion of hFOB1.19 osteoblasts was observed on the surface of the modular plates (Figure 5).

### 3.4. Adhesion and Microbial Biofilm Formation Were Selectively Reduced

Figure 6 shows the macroscopic image of the colonies of microorganisms desorbed from the surface of the titanium–aluminum–vanadium material. Table 2 shows the number of colony-forming units on the surface of the plates.

*L. rhamnosus* and *S. mutans* strains reduced the TTC to 2 (++) and 3 (+++), respectively. The highest degree of TTC reduction—4 (++++)—was observed with *S. epidermidis*. MTT reduction by C. albicans fungi was not detected.

The macroscopic analysis of the degree of TTC and MTT reduction by the investigated strains is shown in Figure 7.

In addition, microCT analysis revealed bacteria on the surface of the titanium alloy plates. Differences in the coverage of bacterial strains were observed (Figure 8). 

### 3.5. Scanning Electron Microscopy (SEM) Revealed Differences in Biofilm Formation

The examinations carried out using scanning electron microscopy showed a biofilm on the surface of the material. The highest degree of adhesion on the surface of the tested materials was detected by the strains of *S. epidermidis* and *S. mutans*. (Figure 9).

## 4. Discussion

This study sought to evaluate the biological properties of Ti-6Al-4V implants designed for osteointegration of the jaw bone (Figure 1, Figure 2, Figure 3, Figure 4 and Figure 5). In general, 3D technology is becoming increasingly popular in bone regeneration. In surgical intervention, 3D models using CAD/CAM techniques have been used in facial reconstruction in plastic surgery [21]. Three-dimensional-printed titanium alloy plates produced by selective laser melting (SLM) have been shown by Lin et al. [22] to be useful in healing acetabular fractures. Recently, some alloys designed from Ti with three-dimensional (3D) reconstruction have been mentioned in the literature, including in a study focused on the reconstruction of mandibular defects by Xue et al. [23]. The usefulness of CAD/CAM techniques in upper maxillary waferless repositioning has also been described by Mazzoni and co-workers [24]. The main advantage pointed out by all authors is the possibility of individual adaptation to the patient’s anatomy, which is known for the osteointegration properties of the material. The osseointegration process is considerably facilitated by the properties of titanium implants, as observed and described by Branemark after the insertion of titanium into bone tissue [5,25]. This process involves the structural and functional connection of the living bone to the surface of the implanted material, which occurs as a result of continuous remodeling of the bone tissue. Bone tissue remodeling includes osteogenesis and resorption processes that are influenced by the biomechanical forces resulting from the physical work of the implant [5,25]. When bone tissue comes into contact with a biomaterial containing a high proportion of titanium, oxidation occurs, leading to the formation of a coating of titanium oxide (TiO_2_) and rutile [26]. This relationship facilitates the formation of a structural bond between the implant and the bone tissue. The pH of titanium oxide is comparable to that of bone tissue, which provides protection against corrosion and acts as a barrier against the external environment [9,27]. Tests with cell lines intended for the evaluation of the biomaterials L929 and Balb/3T3 have shown that the produced modular plates do not exhibit cytotoxicity (Figure 5). Furthermore, our studies show that hFOB1.19 osteoblasts adhere to the surface of the modular plates, confirming the known properties of titanium alloys and suggesting favorable osteointegration properties of the material [25].

An additional consideration when using biomaterials is the potential risk of infection due to the formation of biofilm structures. In this study, the susceptibility of the alloy Ti-6Al-4V to adhesion and biofilm formation by selected microorganisms was investigated. The tests were carried out with reference strains of *S. mutans*, *C. albicans*, *L. rhamnosus* and *S. epidermidis* (Figure 8 and Figure 9).

*S. mutans* is an important cause of dental caries, a disease in which microorganisms form a biofilm on the surface of gums and teeth. The adhesive properties of *S. mutans* promote the colonization of the oral cavity and the formation of a biofilm structure [28]. The same applies to bacteria of the *Lactobacillus* species. *Lactobacillus* spp. are considered to be the most cariogenic bacteria in the oral cavity, as they ferment sugars into acidic products and play an important role in the development of caries. *C. albicans* can be part of the oral microbiota and is involved in opportunistic infections, including oral candidiasis. Infections caused by *C. albicans* often lead to the development of biofilm structures. These structures consist of clusters of hyphae, pseudohyphae and blastoconidia embedded in the extracellular matrix [29]. It is worth noting that the adhesive abilities of these microorganisms are particularly important, especially in prosthetic restorations. The reference strain used, *S. epidermidis* RP62A, has strong adhesive abilities and the ability to form biofilm structures. *S. epidermidis*, which is part of the microbiota of the skin and mucous membranes, is an important factor in infections associated with the use of biomaterials [30,31,32,33].

The quantitative assessment of adhesion and biofilm-forming ability revealed the highest value of colony-forming units for *S. mutans* and the lowest for *C. albicans* (Figure 7). A correlation was found between the two methods when evaluating the adhesion capacity of streptococci. The results obtained indicate a higher adhesion capacity of streptococci compared to other microorganisms. Studies by other authors confirm that *S. mutans* has a high ability to form a biofilm on various materials used in medicine, e.g., those made of zirconium, glass fiber or NiTi alloy, which can lead to its corrosion [34,35,36]. According to Meza-Siccha et al., *S. mutans* has a greater ability to adhere to titanium and zirconium surfaces compared to other oral bacteria such as *Porphyromonas gingivalis* or *Streptococcus sanguinis* [37]. A study conducted in 2020 by Oda et al. showed that streptococci adhere much more strongly to zirconium oxide than to pure titanium. Among the streptococci selected for the test, *S. mutans* showed a lower adhesion capacity than *S. sanguinis*, *Streptococcus gordonii* or *Streptococcus oralis*. The adhesion capacity of *S. mutans* was similar for both biomaterials used [34].

*S. mutans* is considered the primary cariogenic microorganism because it is able to produce significant amounts of glucans and acids that exceed the buffering capacity of saliva. This gives it an advantage over other commensal species in the oral cavity. In addition, *S. mutans* can coaggregate with other oral microorganisms and penetrate deeper into the gingival tissue, leading to the dissolution of hydroxyapatite crystals present in the enamel and dentin, which ultimately favors the development of caries [38].

Kucharíková et al. [39] observed a lower colonization of bacteria and fungi on titanium surfaces modified with antimicrobial compounds compared to unmodified surfaces. The authors of the study used vancomycin for *Staphylococcus aureus* and caspofungin for *C. albicans*. The biofilm formation of S. aureus was inhibited by 99.9% and in the case of C. albicans by 89% compared to the Ti surface without antibiotics.

In addition, the surface properties of the biomaterial, such as roughness or hydrophobicity, can significantly influence the colonization of titanium surfaces by fungi. In a study by Mathieu Mouhat et al., a correlation between surface roughness and colonization with *C. albicans* was demonstrated. C. albicans formed a biofilm on rougher titanium surfaces to a greater extent than on smoother or moderately rough surfaces [37].

The results indicate that the adhesion of *S. mutans* and *S. epidermidis* to the material surface is higher, while the adhesion of *C. albicans* fungi is reduced. This suggests that further surface functionalization is required to reduce the properties that promote bacterial biofilm formation. Surface modification of alloys can be achieved, for example, by coating implanted materials with a layer that exhibits controlled antibiotic release or by coating with SiO_2_-TiO_2_ layers that help to reduce the formation of a biofilm [38,39]. Hydrophilic coatings based on polyethylene glycol, polyethylene oxide or hyaluronic acid can minimize bacterial adhesion [39]. However, to promote osteoblast adhesion, the surface must be modified. Although hyaluronic acid limits bacterial adhesion, it also inhibits mammalian cell adhesion. The chitosan present on the titanium surface can have a bactericidal effect and strengthen the functions of osteoblasts [39].

The test results confirmed the identity of the fixations of the mandibular bone fragments with modular and standard miniplates. The use of modular plates with multiple elements in osteosynthesis of the mandibular angle resulted in better axial stabilization compared to standard methods, proving their validity. Titanium alloys have been used in dentistry since around 1981, with Ti-6Al-4V being the most commonly used alloy. Currently, research is taking a novel approach by using 3D SLM printing technology to produce biocompatible implants with personalized shapes. However, it has become apparent that it is also necessary to develop a functional surface that can reduce the formation of a bacterial biofilm.

In conclusion, the investigated modular titanium plates showed no cytotoxic effects. In addition, the material surface showed adhesion of osteoblasts. The surface of the plates showed the highest adhesion capacity for *S. mutans* and *S. epidermidis*, while the fungus *C. albicans* showed the lowest adhesion capacity. Additional surface modification is required to reduce the formation of a bacterial biofilm.

## Figures and Tables

**Figure 1 biomedicines-12-01466-f001:**
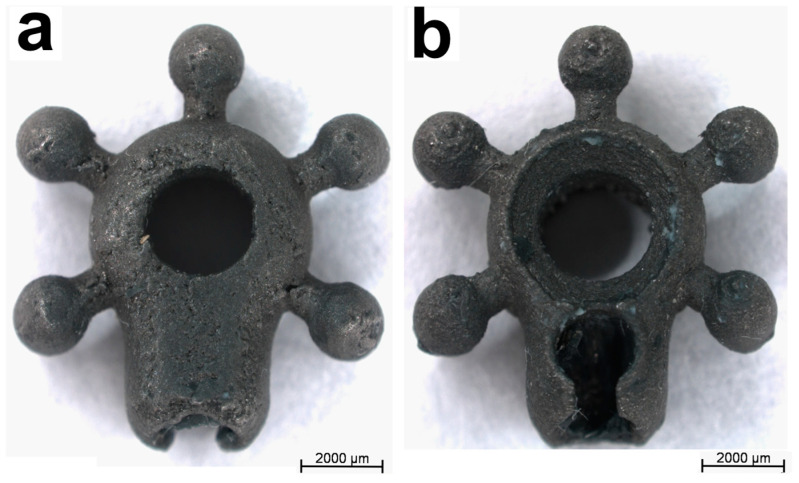
Macroscopic image of modular plates made of Ti6Al4V titanium alloy. (**a**): front, (**b**): back side.

**Figure 2 biomedicines-12-01466-f002:**
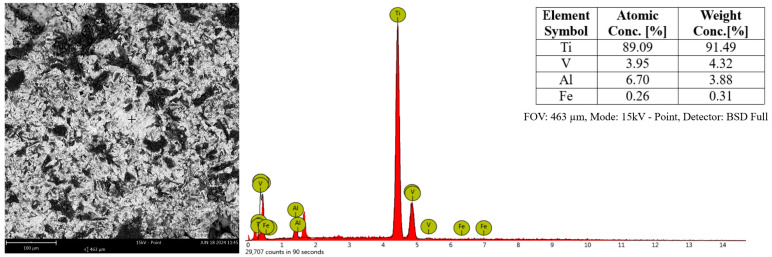
SEM image of a sample of Ti6Al4V, together with the spectrum obtained by the EDS technique and a table of the weight and atomic percentages analysed.

**Figure 3 biomedicines-12-01466-f003:**
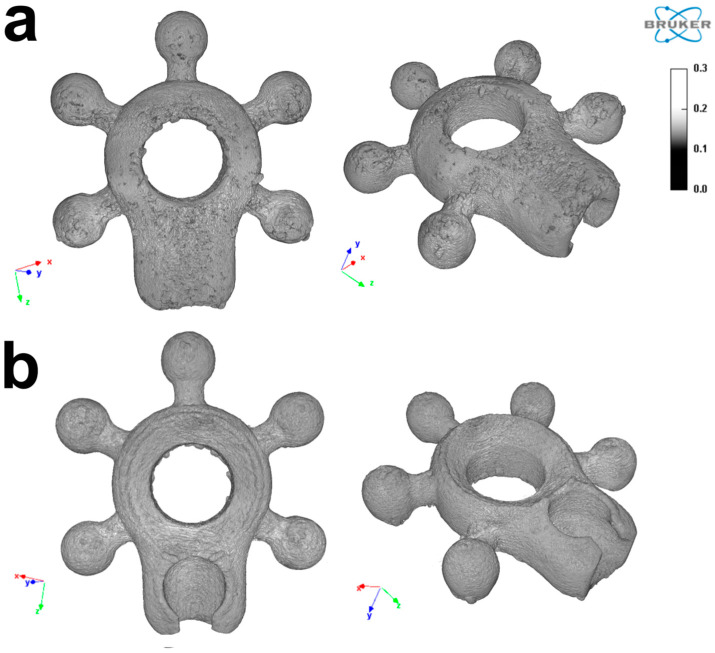
Morphological image of titanium plates using a microcomputer tomography view: (**a**) from the front; (**b**) from the back.

**Figure 4 biomedicines-12-01466-f004:**
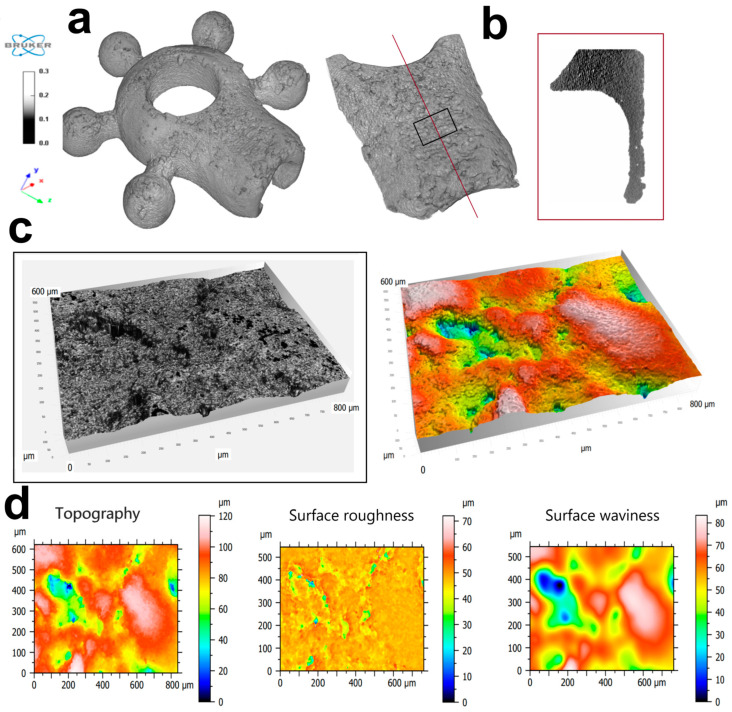
Sample area of the implant frontal plane, using a Leica DCM8 optical profilometer: (**a**) reconstruction of an example plate using SkyScan 1172 µCT, Bruker; (**b**) longitudinal profile of the plate (along the red line), and the area of the roughness analysis (black square); (**c**) sample area for which the 600 × 800 µm roughness parameters were measured obtained with the confocal microscope and the image from the Leica Map software; (**d**) colour maps showing the topography, roughness and waviness of the surface under examination.

**Figure 5 biomedicines-12-01466-f005:**
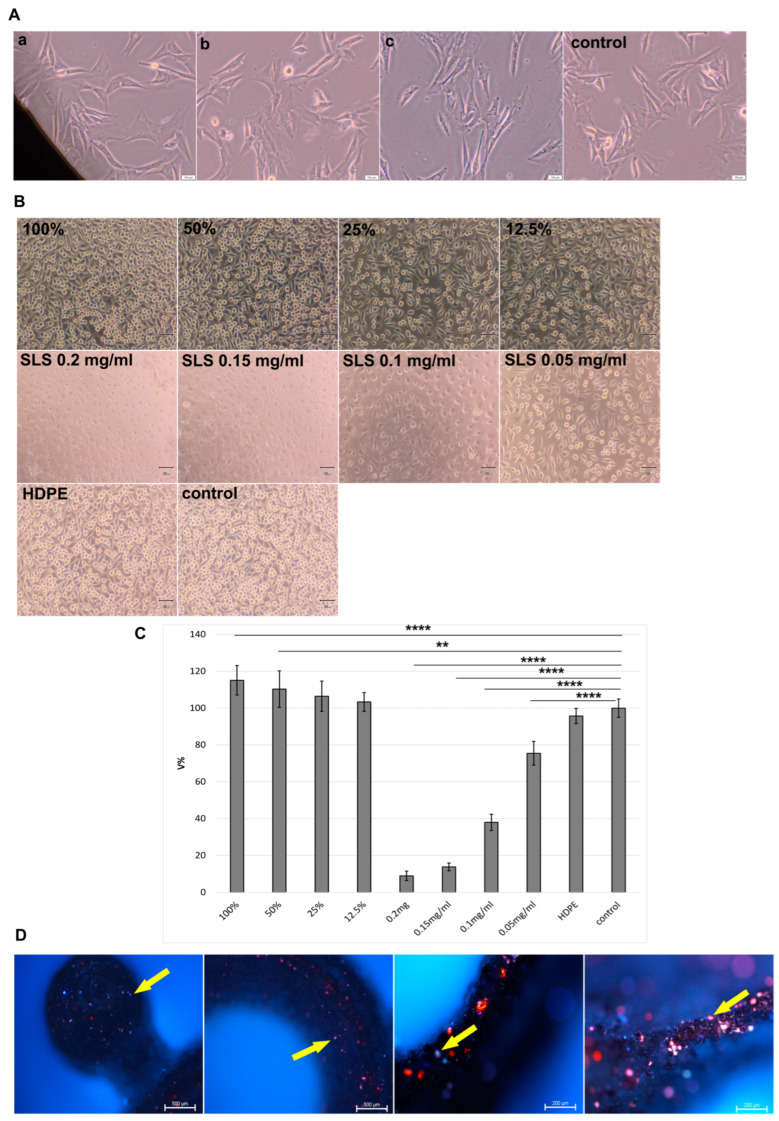
Evaluation of the modular plates in cell cultures: (**A**) direct contact of the material with Balb/3T3 cells; cell morphology after 24 h with the material (**a**) at the edge of the sample, (**b**) near the sample, (**c**) in the remaining part of the well, and control culture without contact with the tested material (100× magnification); (**B**) morphology of L929 cells after 24 h contact with extracts from the tested material (100%, 50%, 25%, 12.5%); positive control: SLS solutions (0.2–0.05 mg/mL); negative control: HDPE (100% high density polyethylene extract); control: blank (100× magnification); (**C**) MTT test after 24 h of contact of L929 cells with the tested extracts: 100% vs. Control: *p* = 0.000013, 50% vs. Control: *p* = 0.0024, SLS extracts vs. control: *p* = 0.000012; one-way ANOVA, post hoc Tukey test, ** *p* < 0.01, **** *p* < 0.0001; (**D**) hFOB1.19 osteoblast adhesion; yellow arrows indicate cells on the surface of the material; DAPI/propidium iodide labeling.

**Figure 6 biomedicines-12-01466-f006:**
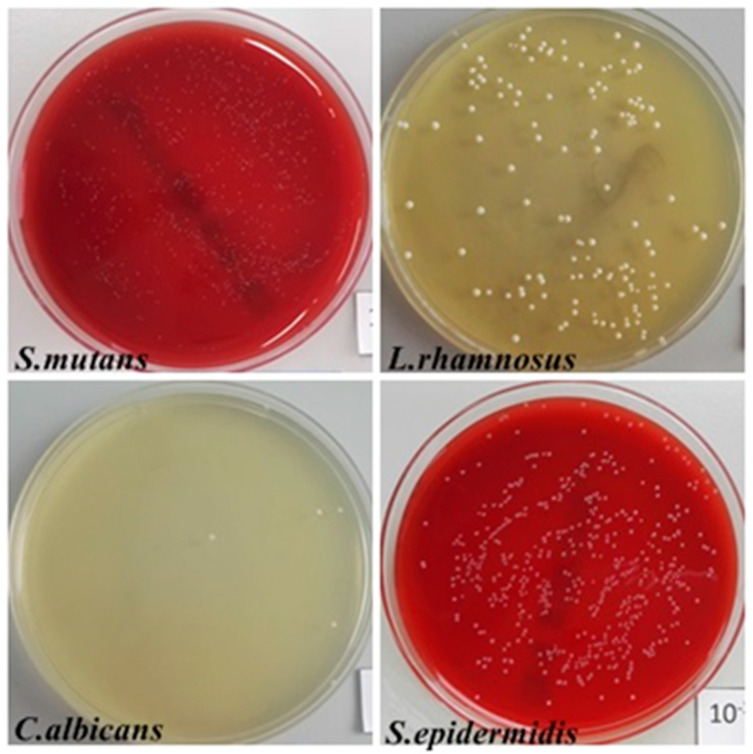
Macroscopic image of colonies of streptococci (*S. mutans*, *S. epidermidis*), fungi (*C. albicans*) and bacilli (*L. rhamnosus*) desorbed from the surface of the tested material under the influence of saponin.

**Figure 7 biomedicines-12-01466-f007:**
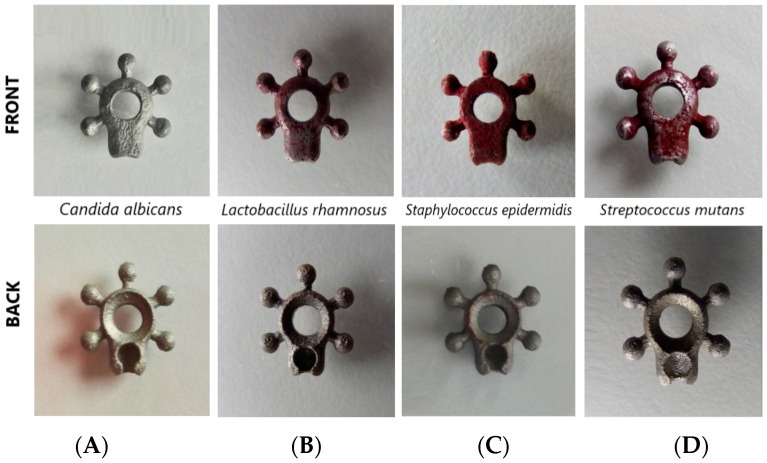
Qualitative assessment of the degree of TTC reduction by (**A**) fungi *Candida albicans* and bacteria: (**B**) *Lactobacillus rhamnosus*; (**C**) *Staphylococcus epidermidis*; (**D**) *Streptococcus mutans*.

**Figure 8 biomedicines-12-01466-f008:**
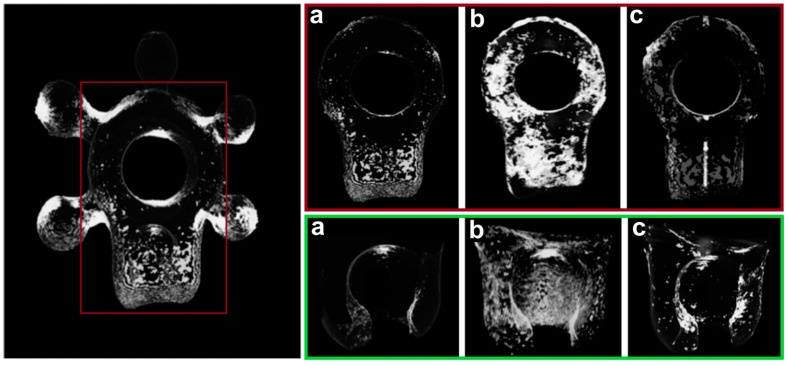
Plate surface (front and back side) obtained by differential analysis in microCT images as a result of differences before and after application of different bacteria reconstructions: (**a**) *Lactobacillus rhamnosus*; (**b**) *Staphylococcus epidermidis*; (**c**) *Streptococcus mutans*.

**Figure 9 biomedicines-12-01466-f009:**
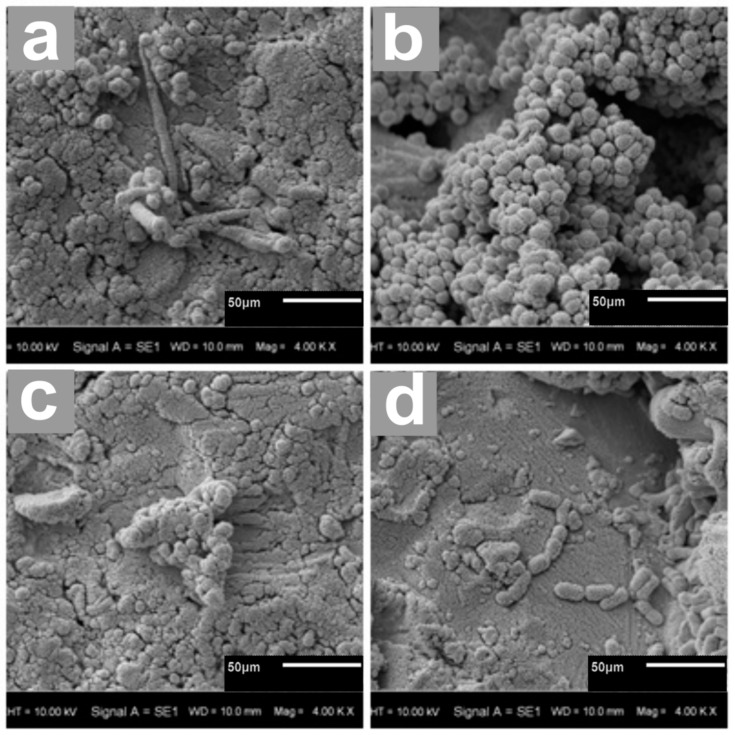
The biofilm-covered surface of modular plates under a scanning electron microscope (SEM). (**a**) *C. albicans*; (**b**) *S. epidermidis*; (**c**) *S. mutans*; (**d**) *L. rhamnosus*. Magn. 400×.

**Table 1 biomedicines-12-01466-t001:** Values of surface texture parameters determined for titanium implants.

Textural Structure Parameters	Surface Topography	Surface Roughness	Surface Waviness
Sa [µm]	12.587 ± 1.731	2.757 ± 0.863	11.047 ± 1.037
Sq [µm]	16.534 ± 1.947	4.542 ± 1.298	14.410 ± 1.244
Sz [µm]	132.836 ± 17.585	93.320 ± 30.105	88.581 ± 7.371
Ssk [-]	−0.936 ± 0.292	−1.560 ± 1.74	−0.841 ± 0.186
Sku [-]	4.506 ± 0.796	17.908 ± 3.104	4.135 ± 0.173
Sp [µm]	53.920 ± 25.915	46.732 ± 31.245	30.604 ± 3.131
Sv [µm]	78.916 ± 8.33	46.587 ± 1.139	57.977 ± 4.241

**Table 2 biomedicines-12-01466-t002:** Number of colony-forming units per milliliter of suspension (CFU/mL).

Species	*S. mutans* M ± SD	*L. rhamnosus* M ± SD	*C. albicans* M ± SD	*S. epidermidis* M ± SD
CFU/mL	6.38 × 10^6^ ± 2.30 × 10⁵	1.3 × 10^6^ ± 1.20 × 10⁵	6.0 × 10^4^ ± 2.0 × 10^3^	1.07 × 10^6^ ± 7.55 × 10⁴

M—mean value; SD—standard deviation.

## Data Availability

The raw data supporting the conclusions of this article will be made available by the authors on request.

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
