# Peer review of "Three-Dimensional-Printed Modular Titanium Alloy Plates for Osteosynthesis of the Jawbone"

_biomedicines, 2024, doi:10.3390/biomedicines12071466_

Round 1

Reviewer 1 Report

Comments and Suggestions for Authors

In this manuscript, the authors made 3D-printed modular titanium plate and examined the properties. They claimed that there are no harmful effects by the 3D-printed plate. This manuscript has potential to be fascinating paper, though the reviewer felt that it is still developing stage. The followings are comments.

1. adequate control is missing.

The authors would intend to clarify whether 3D-printed plate exhibits harmful effects or not. It can be rephrasing that 3D-printed plate has any advantages as compared to the plate fabricated by cast or milling. Therefore, the reviewer felt that the control should be the use of casted or milling-machine prepared titanium plate.

2. In the abstract and introduction, the authors mentioned titanium alloys are commonly used for medical implants for stabilizing bone fragments. Therefore, the reviewer is curious whether this 3D-printed plates has any advantage for osteoblastic differentiation.

3. In some cases, infection-mediated inflammatory responses induce trouble for metal implant, especially in craniofacial region. Therefore, the reviewer is curious whether this 3D-printed plates has any influence on inflammation and immune responses.

Author Response

In this manuscript, the authors made 3D-printed modular titanium plate and examined the properties. They claimed that there are no harmful effects by the 3D-printed plate. This manuscript has potential to be fascinating paper, though the reviewer felt that it is still developing stage. The followings are comments.

  1. adequate control is missing.

The authors would intend to clarify whether 3D-printed plate exhibits harmful effects or not. It can be rephrasing that 3D-printed plate has any advantages as compared to the plate fabricated by cast or milling. Therefore, the reviewer felt that the control should be the use of casted or milling-machine prepared titanium plate.

Dear Reviewer, thank you for this valuable comment. Indeed, as we described in the discussion section (L440 and following), titanium alloys have been used in dentistry since about 1981, and Ti-6Al-4V is the most commonly used alloy. Properties of Ti and its alloy are commonly known as positive aspects in osteointegration and no cytotoxic effects and control as cast or mill prepared titanium plate not be strongly different in cytotoxic effects. We provide in-depth analysis in trace element contents and physical properties of our titanium alloy. The clue is in different point - in the shape of these alloys. Currently, research is taking a novel approach by using 3D SLM printing technology to produce biocompatible implants with personalised shapes. Our studies have been carried out to design and produce new geometric elements from known Ti alloys. The use of modular plates with multiple elements in mandibular angle osteosynthesis resulted in better axial stabilisation compared to standard methods, proving their validity. Our modular titanium plates showed no cytotoxic effects and the material surface showed adhesion of osteoblasts, however additional surface modification is required to reduce the formation of bacterial biofilm and this is the next step in 3D SLM printing technology for us: surface functionalisation. We hope that this explanation will be sufficient for you.

  1. In the abstract and introduction, the authors mentioned titanium alloys are commonly used for medical implants for stabilizing bone fragments. Therefore, the reviewer is curious whether this 3D-printed plates has any advantage for osteoblastic differentiation.

 Dear Reviewer, thank you for this valuable comment. In general, mesenchymal stem cell differentiation and osteoblast maturation require some biological factors. In the tissue microenvironment, this process occurs naturally and also some biomaterials can be modified to enhance this process. One of the most recognised differentiation factors used in the functionalisation of biomaterials to enhance the differentiation of mesenchymal stem cells into osteoblasts is hydroxyapatite (Hap). Our titanium alloy plates have not been included Hap, hovewer, 3D printed plates can, after osteointegration, participate in osteoblastic differentiation on their surface in a passive manner. Functionalisation of our 3D plate is an interesting option for future research, we will think about this possibility. We hope that this explanation will be sufficient for you.

  1. In some cases, infection-mediated inflammatory responses induce trouble for metal implant, especially in craniofacial region. Therefore, the reviewer is curious whether this 3D-printed plates has any influence on inflammation and immune responses.

Dear Reviewer, thank you for your valuable comment. The immune response generated by these 3D printed plates may occur due to infection. In fact, all implants can cause this type of problem. In our studies, we evaluate biofilm formation on 3D titanium alloy as biofilm formation on implants is a real big problem in implantation. The results obtained showed that the surface of this material had the highest adhesion capacity for S. mutans and S. epider-midis, while the fungus C. albicans showed the lowest adhesion capacity. These properties suggest that additional surface modification is required to reduce the formation of a bacterial biofilm. In order to fully investigate whether these 3D printed plates have any influence on inflammation and immune responses, in vivo implantation is required. In the near future, additional modification of the surface of this 3D titanium alloy to reduce biomaterial formation seems to be a good idea to conduct more in-depth research with this material. We hope that this explanation will be sufficient for you.

Reviewer 2 Report

Comments and Suggestions for Authors

The manuscript is interesting and novel since the authors used the new technology  3D SLM printing to produce their implant. Although more test with osteoblasts can be done to show the bioactivity of their material as bone implant the results are enough to support a publication. It should be published after minor revision

Comments:

Title: The authors used a Titanium alloy and not pure Titanium,  the title should be changed to  “Titanium Alloy Plates” or “Ti-6Al-4V Plates”

Table 1: How was the elemental composition measured?

Show the XRD (X-ray diffraction) pattern for the Ti-6Al-4V alloy

Paragraph 3.1: Add a schematic description of the measured surface texture parameters

Author Response

Title: The authors used a Titanium alloy and not pure Titanium,  the title should be changed to  “Titanium Alloy Plates” or “Ti-6Al-4V Plates”

Dear Reviewer, thank you for your comment. We have changed the title according to your suggestion.

Table 1: How was the elemental composition measured?

Show the XRD (X-ray diffraction) pattern for the Ti-6Al-4V alloy

Unfortunately, it has been a long time since this measurement and currently, due to the failure of the XRD equipment, we are unable to recover the full spectrum from the XRD. Therefore, in order to validate the results published in this paper, an EDS study was carried out based on the SEM measurement. Scanning electron microscopy (SEM) coupled with energy-dispersive X-ray diffraction (EDS)(Phenom ProX; Thermo Scientific, Waltham, MA, USA) was used. The elemental composition as evaluated by EDS was analyzed on weight and atomic percentage base for all samples. We added Figure 2 to manuscript with SEM image of a sample of Ti6Al4V, together with the spectrum obtained by EDS technique and a table of the weight and atomic percentages analysed

Reviewer 3 Report

Comments and Suggestions for Authors

Authors have written 3D-Printed Modular Titanium Plates for Osteosynthesis of the Jawbone with cells and bacteria studies. The works are new and have good quality of images.

Please modify image labelling style and citations

Figure 1-8 are not cited in the discussion. please include them. Please connect your result and compare with literature in the discussion.

Figures should be labelled as a, b, c, d to replace A, B, C, D.

Figure 3

Please increase the font of letter in Panel A left top and panel B

Figure 4 

Please increase the font of letters of scale bar and bars in Panel A ,B,D

Line 324

Figure 5 figure legend should be 

Figure 5. Macroscopic image of colonies of streptococci, fungi and bacilli desorbed from the surface of the tested material under the influence of saponin.

Figure 8

Please include scale bar in the image.

 L. rhamnosus

Please use italic in Line 215 and line 328

S. mutans 

Please use italic in Line 32,215,328,369,396,398

S. epidermidis

Please use italic in Line 32, 329

C. albicans

Please use italic in Line 33, 329, 412.

Line 419

SiO2-TiO2 should be SiO2-TiO2  

Line 439

Author Contributions: Conceptualization: M.S., M.D.; Methodology: P.K., M.P., J.N., A.N., A.R.; Investigation: A.R., M.P., A.N., P.K.; Resources: MW; Validation: P.K, A.R., A.N., M.P., J.N.; Visualization: P.K., M.P., J.N., A.N., A.R.; Formal Analysis: M.S., M.P., M.W., M.D.; Writing – Original Draft: A.R., M.P., J.N.; Writing – Review & Editing: M.S., J.N., M.D., A.N., A.R.; Supervision: A.R.; Project Administration: M.S.; Funding Acquisition: M.S. 

Please follow MDPI Reference style 

For example first reference

Smith, K.E.; Dupont, K.M.; Safranski, D.L., Up to five authors (2016) Use of 3D printed bone plate in novel technique to surgically correct hallux valgus deformities. In: Techniques in Orthopaedics. Lippincott Williams and Wilkins, pp 181–189

Author Response

Comments and Suggestions for Authors

Authors have written 3D-Printed Modular Titanium Plates for Osteosynthesis of the Jawbone with cells and bacteria studies. The works are new and have good quality of images.

Please modify image labelling style and citations

Dear Reviewer, thank you for your comments. We agree. We changed the image labelling and citations according to your suggestions.

Figure 1-8 are not cited in the discussion. please include them. Please connect your result and compare with literature in the discussion.

Dear Reviewer, thank you for your comments. We have included figures in the discussion. In the discussion, we compared our results with those of other authors. We have made this section clearer according to your suggestion. We hope this is satisfactory.

Figures should be labelled as a, b, c, d to replace A, B, C, D.

Dear Reviewer, thank you for this comment, we fully agree wit this. We corrected figures labbeling acccording to this.

Figure 3

Please increase the font of letter in Panel A left top and panel B

Dear Reviewer, thank you for this comment. We increased the front of letter on this image.

Figure 4

Please increase the font of letters of scale bar and bars in Panel A ,B,D

Dear Reviewer, thank you for this comment, we corrected Figure 4 according to your suggestion.

Line 324

Figure 5 figure legend should be

Dear Reviewer, thank you for your comment. We fully agree with this statement. We added a description under the  Figure 5.

Figure 5. Macroscopic image of colonies of streptococci, fungi and bacilli desorbed from the surface of the tested material under the influence of saponin.

Figure 8

Please include scale bar in the image.

Dear Reviewer, thank you for this comment, we introduced scale bar on Figure 8.

  1. rhamnosus

Please use italic in Line 215 and line 328

  1. mutans

Please use italic in Line 32,215,328,369,396,398

  1. epidermidis

Please use italic in Line 32, 329

  1. albicans

Please use italic in Line 33, 329, 412.

Line 419

SiO2-TiO2 should be SiO2-TiO2

Dear Reviewer, thank you for this comments, we used italic in this all above mentioned places and also correct SiO2-TiO2 to appropriate font and also another similar errors like in CO2.

Line 439

Author Contributions: Conceptualization: M.S., M.D.; Methodology: P.K., M.P., J.N., A.N., A.R.; Investigation: A.R., M.P., A.N., P.K.; Resources: MW; Validation: P.K, A.R., A.N., M.P., J.N.; Visualization: P.K., M.P., J.N., A.N., A.R.; Formal Analysis: M.S., M.P., M.W., M.D.; Writing – Original Draft: A.R., M.P., J.N.; Writing – Review & Editing: M.S., J.N., M.D., A.N., A.R.; Supervision: A.R.; Project Administration: M.S.; Funding Acquisition: M.S.

Dear Reviewer, thank you for your comment, we corrected author contribution form. We hope this is satisfactory for you.

Please follow MDPI Reference style

For example first reference

Smith, K.E.; Dupont, K.M.; Safranski, D.L., Up to five authors (2016) Use of 3D printed bone plate in novel technique to surgically correct hallux valgus deformities. In: Techniques in Orthopaedics. Lippincott Williams and Wilkins, pp 181–189

Dear Reviewer, thank you for this valuable comment, we provided correction of References style. Addditionally, according to Editor comment, we reduce our self-citation to 20% and therefore another changes in References occured, becouse we removed our two positions from references. We also provide additionall references according to discussion section modyfications. We hope that this changes will be satisfactory for you.

Round 2

Reviewer 1 Report

Comments and Suggestions for Authors

In this revised manuscript and response letter, the authors addressed to the reviewer's comments adequately and explained the situation that the authors intend to inform the possibility of 3D-printed metal for implant. As the authors explained in the response letter, this manuscript lack some data the reviewer requested. However, to consider the authors' intend to inform the possibility of 3D-printed metal for implant, it is understandable to publish at this stage.

Author Response

Deaqr Reviewer, 

We would like to thank you for your review and for all of your valuable comments.

Reviewer 3 Report

Comments and Suggestions for Authors

Authors have improved manuscript. 

Figure  2 please include scale bar in the left image

Figure 5 d

please show the right panel of figure 5d

Please include the figure 5 legend at bottom of figure 5 at same page by reducing number of words.

Some of content of figure legend can be moved into result after the figure 5. 

Table 1

please use MDPI style for table

There are top line, second line and bottom line. Please remove the line 3-9 and all of vertical lines. Please move Mean ± SD after Table 1. Values of surface texture parameters determined for titanium implants.

Line 50

Please change [2][3] to[2,3]

line 63

Please change [1][2][6][3] to [1-3,6]

LIne 67

Please change  [7][8] to[7,8]

line 72 

 Please change [7][6] to [6,7]

Line 77

 Please change [9][8][4, 8] to [4,8,9]

Line 79-89

please include reference citation 

line 94

Please change  [10][16] to [10,16].

line 101

please remove the empty line 

line 124

same as above

Line 171

[13][14][15].

Line 203 

Please change [12][11] to [11,12]

line 217

Please change[13][14] to [13,14]

Line 263

Please change   [15][12] to [12,15]

Line 454 

Please change [35][37] to [35,37]

Author Response

Authors have improved manuscript. 

Figure  2 please include scale bar in the left image

 Dear Reviewer, thank you for this comment, then scale bar are visible on the left bottom on SEM image.

Figure 5 d

please show the right panel of figure 5d

Dear Reviewer, thank you for this comment, we made correction and now Figure 5 is visible in the manuscript.

Please include the figure 5 legend at bottom of figure 5 at same page by reducing number of words.

Dear Reviewer, thank you for this comment, we moved the legend from Figure 5 according to your suggestion. Plese note, that this version of manuscript is track changes version and after removing corrected places, manuscript will have final look and number of pages will be decreasing.

Some of content of figure legend can be moved into result after the figure 5. 

Dear Reviewer, thank you for this comment, we moved some information from legend of Figure 5 to results. All changes are marked in the manuscript.

Table 1

please use MDPI style for table

There are top line, second line and bottom line. Please remove the line 3-9 and all of vertical lines. Please move Mean ± SD after Table 1. Values of surface texture parameters determined for titanium implants.

Dear Reviewer, thank you for this comment, we corrected Table 1 according to yours suggestions and also Table 2.

Line 50

Please change [2][3] to[2,3]

line 63

Please change [1][2][6][3] to [1-3,6]

LIne 67

Please change  [7][8] to[7,8]

line 72 

 Please change [7][6] to [6,7]

Line 77

 Please change [9][8][4, 8] to [4,8,9]

Line 79-89

please include reference citation 

line 94

Please change  [10][16] to [10,16].

line 101

please remove the empty line 

line 124

same as above

Line 171

[13][14][15].

Line 203 

Please change [12][11] to [11,12]

line 217

Please change[13][14] to [13,14]

Line 263

Please change   [15][12] to [12,15]

Line 454 

Please change [35][37] to [35,37]

Dear Reviewer, thank you for this comment, we corrected citation according to this suggestion in all this places. We hope that all of the corrections made to this manuscript will be satisfactory to you.